# Activation of NLRP3 Is Required for a Functional and Beneficial Microglia Response after Brain Trauma

**DOI:** 10.3390/pharmaceutics14081550

**Published:** 2022-07-26

**Authors:** Ana Belen Lopez-Rodriguez, Céline Decouty-Perez, Victor Farré-Alins, Alejandra Palomino-Antolín, Paloma Narros-Fernández, Javier Egea

**Affiliations:** Molecular Neuroinflammation and Neuronal Plasticity Research Laboratory, Hospital Universitario Santa Cristina, Instituto de Investigación Sanitaria-Hospital Universitario de la Princesa, 28009 Madrid, Spain; celinedecouty96@gmail.com (C.D.-P.); victorfarre@hotmail.com (V.F.-A.); alejandra.palominoantolin@gmail.com (A.P.-A.); palomanf22@gmail.com (P.N.-F.)

**Keywords:** traumatic brain injury, inflammasome, NLRP3, neuroinflammation, microglia, astrocytes, MCC950

## Abstract

Despite the numerous research studies on traumatic brain injury (TBI), many physiopathologic mechanisms remain unknown. TBI is a complex process, in which neuroinflammation and glial cells play an important role in exerting a functional immune and damage-repair response. The activation of the NLRP3 inflammasome is one of the first steps to initiate neuroinflammation and so its regulation is essential. Using a closed-head injury model and a pharmacological (MCC950; 3 mg/kg, pre- and post-injury) and genetical approach (NLRP3 knockout (KO) mice), we defined the transcriptional and behavioral profiles 24 h after TBI. Wild-type (WT) mice showed a strong pro-inflammatory response, with increased expression of inflammasome components, microglia and astrocytes markers, and cytokines. There was no difference in the IL1β production between WT and KO, nor compensatory mechanisms of other inflammasomes. However, some microglia and astrocyte markers were overexpressed in KO mice, resulting in an exacerbated cytokine expression. Pretreatment with MCC950 replicated the behavioral and blood–brain barrier results observed in KO mice and its administration 1 h after the lesion improved the damage. These findings highlight the importance of NLRP3 time-dependent activation and its role in the fine regulation of glial response.

## 1. Introduction

Traumatic brain injury (TBI) occurs when a sudden blunt impact, skull penetration, or explosive blast damages the brain. It is one of the major causes of mortality and disability worldwide and has a complex pathophysiology that can be divided into primary and secondary injuries [1]. Primary damage is the consequence of the external forces applied directly to the brain, which produce mechanical damage to nervous tissue, such as contusion and hemorrhage. Delayed secondary injury includes complex pathophysiological events, such as glutamate excitotoxicity, oxidative stress, and inflammatory response, that occur weeks to months after brain injury [2,3]. 

Neuroinflammation is the innate immune response of the central nervous system (CNS), which removes brain infections and repairs tissue damage. It is a complex and well-coordinated process, involving different groups of CNS glial cells and peripheral immune cells, essential for CNS regeneration, but when this response is dysregulated, it leads to chronic neuroinflammation, which is detrimental to the brain [4]. Microglia, the resident innate immune cells in the CNS, have phagocytic and antigen-presenting activity and contribute to normal brain development, function, and repair [5]. Under pathological conditions, they are rapidly activated and initiate various inflammatory cascades to restore tissue homeostasis. In the context of TBI, microglia activation determines the fate of activated astrocytes, which, at the same time, release a variety of regulators that control the microglial phenotype and its cellular function [6]. In this way, microglia and astrocytes establish a bidirectional interaction, essential for the resolution or chronification of inflammation.

The activation of different inflammasome complexes is a fundamental step at this stage of neuroinflammation [7]. The inflammasomes are cytosolic multiprotein complexes that assemble and activate pro-inflammatory caspase-1, which is involved in the maturation and secretion of the inflammatory cytokines IL1β and IL-18, inducing pyroptosis of inflammatory cells [8]. Inflammasomes are distributed differently in the brain: NOD-like receptor family, pyrin domain containing 3 (NLRP3) is predominantly present in microglia, but is also present in oligodendrocytes and in astrocytes [9,10]. However, NLRP1 and Absent In Melanoma 2 (AIM2) are expressed in neurons [11]. In recent years, researchers have focused on the role of the NLRP3 inflammasome in several brain disorders, including neurodegenerative diseases, such as Alzheimer’s disease or Parkinson’s disease, TBI, and CNS infections. The activation of the NLRP3 inflammasome seems to be responsible for the adverse consequences of CNS injury, and selective pharmacological inhibition of NLRP3 has been shown to be protective in different TBI models [12,13]. However, we recently showed that NLRP3 plays a dual time-dependent role in brain injury, using an experimental ischemia model. NLRP3 activation appears to be important for inducing a proper inflammatory response, as NLRP3 KO animal mortality increases after CNS injury, but the selective inhibition of NLRP3 activation using MCC950 1 h after the ischemia is protective [14]. Nevertheless, the role of the NLRP3 inflammasome in CNS trauma remains complex and controversial and requires further investigation. 

Here, we aimed to contribute to the understanding of the role of NLRP3 in microglial–astrocyte crosstalk in brain trauma by using both genetic and pharmacological approaches to inhibit NLRP3. Our results indicate an important role for NLRP3 inflammasome activation after brain trauma. In NLRP3 KO animals, the post-TBI inflammatory response is exaggerated, with increased reactive astrocyte markers and astrocyte-related cytokines, probably due to a dysfunctional interaction between microglia and astrocytes. In addition, NLRP3 genetic deficiency or pharmacological inhibition prior to brain trauma are detrimental and play an important role in blood–brain barrier (BBB) integrity and neurological damage. These results support the importance of the NLRP3 inflammasome to perform a functional and appropriate microglial response after brain injury that benefits BBB integrity and trauma severity and suggest that NLRP3 activation might also be affecting the function of astrocytes.

## 2. Materials and Methods

### 2.1. Animals

C57Bl6/J (RRID: IMSR_JAX:000664) and NLRP3 -/- (RRID: MGI:5468973) mice were used at approximately 12 weeks old and came from the inhouse colony of the animal facilities of Universidad Autónoma de Madrid (Madrid, Spain). Animals were housed in cages of four at 21 °C with a 12 h light/dark cycle. Food and water access was *ad libitum*. All animal experimentation was performed under the license PROEX 109/18 granted by the Ethics Committee of Universidad Autónoma de Madrid (Madrid, Spain) and in compliance with the Cruelty to Animals Act, 1876, and the European Community Directive, 86/609/EEC. Every effort was made to minimize stress to the animals.

### 2.2. Traumatic Brain Injury Model

Mice were randomly assigned to the Naïve (no lesion) or TBI (lesion) group before the experiment. Those subjected to the closed-head injury (CHI) model, adapted from Flierl et al. 2009 [15], were anesthetized with inhaled isoflurane before the lesion. The TBI was induced by dropping a 50 g weight onto the right hemisphere from a height of 34 cm along a stainless-steel rod. After trauma, mice were closely monitored in an individual cage and oxygen was administered, when necessary, until regular breath was restored. This brain trauma model produces a strong inflammatory response in the right hemisphere, functional and neurological deficits, and a 5–15% mortality rate within the first 5 min after the impact [16,17,18].

### 2.3. Neurological Severity Score (NSS) Test

The neurological severity score (NSS) of the injury is based on the assessment of motor and neurobehavioral functions at 1 and 24 h after trauma, using a modified version of Flierl et al. (2009) [15]. One hour after TBI the functional outcome was assessed by a person that was blind to the experimental groups and treatments and consisted of the evaluation of the 10 parameters described in Table 1. Animals with a score of 10 points at the 1 h test were excluded as it was considered as a humane endpoint criterion. After the 1 h NSS test, animals were returned to their home cages until the next day when the 24 h NSS was performed.

### 2.4. Treatments

The NLRP3 inflammasome inhibitor MCC950 (CP-456773, Sigma-Aldrich, Madrid, Spain) was injected intraperitoneally (i.p.) at 3 mg/kg 30 min prior to the TBI model (pre) to mimic the NLRP3-KO response or one hour after the trauma (post), after all the 1 h behavioral tests were finished. Saline 0.9% i.p. (10 mL/kg) was used as vehicle (vh) for control groups.

### 2.5. Blood–Brain Barrier Integrity Assessment

To evaluate the integrity of the blood–brain barrier, 2% Evans Blue tracer (Sigma-Aldrich Cat# E2129) diluted in saline was injected (3 mL/Kg; i.p.) following the 1 h NSS test. Animals were returned to their home cages and were sacrificed 24 h after trauma by cervical dislocation. The brains were gently removed and sectioned in 2 mm slices using a mouse brain slicer. Four slices were taken and photographed per animal to measure the total area of Evans Blue extravasation. Each slice was individually analyzed using ImageJ 1.52a (ImageJ software, National Institutes of Health, Stapleton, NY, USA) and the summary was represented as total extravasation cubic (mm^3^).

### 2.6. Tissue Preparation

For the analyses of transcriptional changes, animals were terminally anesthetized with a mix of ketamin:xylacin 1:2 (Ketolar 50 mg/mL, Pfizer; Xilagesic 20 mg/mL, Calier Labs, Barcelona, Spain) 24 h after TBI and rapidly transcardially perfused with heparinized saline. Brains were gently removed and a punch of the right (ipsilateral) and left (contralateral) hemispheres was snap frozen and stored at −80 °C until use.

### 2.7. RNA and Protein Extraction

RNA and protein were obtained by double-extraction protocol with Trizol reagent (TRI Reagent, Molecular Research Center, Inc., Cincinnati, OH, USA) according to the manufacturer’s instructions. We proceeded to the phase separation, keeping the phenol phase for protein isolation and the aqueous phase for RNA extraction.

### 2.8. Quantitative Real-Time Polymerase Chain Reaction (qRT-PCR)

After RNA isolation, the RNA yield and quality of each sample were quantified based on optical density using the NanoDrop 2000c spectrophotometer (Thermo Scientific, Waltham, MA, USA). cDNA synthesis was carried out using a High-Capacity iScript cDNA Synthesis Kit (BIO RAD, #1708890; Hercules, CA, USA). Primer and probe sets were designed using NCBI Nucleotide tool and their sequences are shown in Table 2. Samples for qRT-PCR were run in duplicate using SYBR green dsDNA-intercalating fluorescent dye (TB Green Premix, Takara #RR420L, Kusatsu, Japan) or FAM-labelled probes (TaqMan Fast Advanced Master Mix, Applied Biosystems #A44360, Waltham, MA, USA) in a QuantStudio5 Real-Time PCR system (Applied Biosystems) under the cycling conditions: 95 °C for 10 min followed by 95 °C for 10 secs and 60 °C for 30 s for 40–45 cycles. Quantification was achieved by exploiting the delta Ct method using WT-Naïve values to normalize the Ct values. Gene expression data were normalized to the housekeeping gene 18S and expressed as relative concentration.

### 2.9. Western Blot

After protein isolation, the sample yield was quantified using the BCA method (Pierce BCA protein assay kit, #23227; Thermo Scientific) and then equalized to 30 μg per well. Solubilized proteins were resolved by 15% SDS–PAGE at room temperature and then transferred to Immobilon-P membranes (Millipore Corp., Billerica, MA, USA). Membranes were blocked with 4% BSA and then incubated with the primary antibody for IL1β [1:500] (AF-401-NA; R&D systems, Minneapolis, MN, USA) overnight at 4 °C. Membranes were incubated for 1 h at room temperature with the appropriate secondary antibody [1:5000] and developed with Immobilon Crescendo Western HRP Substrate (Millipore; Burlington, MA, USA).

### 2.10. Statistical Analyses

For multiple comparisons, one or two-way analysis of variance (ANOVA) was used with factors being genotype (wild type (WT) or NLRP3 knockout (KO)); lesion (naïve (N), contralateral (C) or ipsilateral (I)); time (1 h or 24 h) and treatment (vehicle (Vh), MCC950 pre-treatment (Pre) or MCC950 post-treatment (Post)). Subsequent post hoc comparisons (Bonferroni’s test) were performed with a level of significance set at *p* ≤ 0.05. Data were not always normally distributed and, in these cases, nonparametric tests were used (Kruskal–Wallis and post hoc pair-wise comparisons with Mann–Whitney U-test). For two-groups comparisons, unpaired t-tests were used when data were normally distributed and the Mann–Whitney U-test was run if data did not pass the assumptions for normality. Data are presented as mean ± standard error of the mean (SEM). Symbols in the graphs denote post hoc tests. Statistical analyses were carried out with the SPSS 22.0 software package (SPSS, Inc., Chicago, IL, USA).

## 3. Results

### 3.1. Inflammatory Profile in WT Mice 24 h after TBI

Traumatic brain injury induced a potent inflammatory response in WT mice 24 h after the lesion. The ipsilateral side showed increased mRNA levels of Nlrp3 and Aim2 inflammasomes, whereas Nlrc4, the adaptor protein ASC, or the caspases 1 and 11 did not change. TBI also increased Il1b mRNA levels, while Il18 remained unchanged. Microglia showed a pro-inflammatory status in the ipsilateral side, with similar but greater changes in astrocytes and pro-inflammatory cytokines. Changes in different inflammasomes are represented in Figure 1a. One-way ANOVA showed a significant effect of the lesion (naïve, contra, ipsi) in Nlrp3 mRNA levels (F(2, 14) = 5.458; *p* = 0.0177); Bonferroni post hoc test confirmed a significant increase on the ipsilateral side compared with naïve animals (*p* = 0.0241) and a trend when compared with the contralateral side (*p* = 0.075). Nlrc4 inflammasome did not show changes, as expected, whereas one-way ANOVA showed a significant effect of the lesion in Aim2 (F(2, 15) = 4.046; *p* = 0.0393); the Bonferroni post hoc test revealed a significant increase on the ipsilateral side compared with naïve animals (*p* = 0.0398). To further analyze the inflammasome components, the adaptor protein ASC (Figure 1b) and the caspases 1 and 11 (Figure 1c) were included, without finding any significant change at 24 h. As “inflammasome products”, Il1b and Il18 were assessed (Figure 1d). Kruskal–Wallis test showed a significant effect of lesion in Il1b mRNA levels (*p* = 0.004), which was confirmed by Mann–Whitney test and revealed a significant increase in the contralateral (*p* = 0.037) and ipsilateral side (*p* = 0.004) in comparison with the naïve group, as well as a difference between the two sides of the injured brain (*p* = 0.037). However, no significant changes were found for Il18.

Microglial activation status is shown in Figure 1e,e′. While Cd11c did not show any change at 24 h after TBI, other reactivity microglia markers presented increases in the ipsilateral side of the lesion. Kruskal–Wallis revealed a significant effect of lesion in Cd11b levels (*p* = 0.035); the Mann–Whitney test confirmed a significant increase in the ipsilateral side in comparison with naïve mice (*p* = 0.004). One-way ANOVA showed an effect of lesion on Clec7a mRNA levels (F(2, 15) = 4.584; *p* = 0.0279) that was confirmed by Bonferroni test, comparing the ipsilateral side and the naïve group (*p* = 0.0387). For Iba1 mRNA levels, one-way ANOVA showed an effect of lesion (F(2, 15) = 4.938; *p* = 0.0225) and Bonferroni test revealed a significant increase in the ipsilateral side when compared with naïve (*p* = 0.0433) and contralateral side (*p* = 0.0450). Finally, Kruskal–Wallis showed a significant effect of lesion on Tyrobp mRNA expression (*p* = 0.002); a subsequent Mann–Whitney test confirmed a significant difference in the contralateral (*p* = 0.004) and the ipsilateral (*p* = 0.004) with the naïve group. Regarding astrocyte activation, different reactivity-associated transcripts were analyzed and are shown in Figure 1f. One-way ANOVA showed a significant effect of lesion on Gfap mRNA expression (F(2, 15) = 26.07; *p* < 0.0001) that was confirmed by Bonferroni test, showing an increase in the ipsilateral side when compared with the naïve (*p* < 0.0001) and the contralateral side (*p* = 0.0004). Steap4 presented a similar activation pattern, one-way ANOVA revealed a significant effect of lesion (F(2, 13) = 17.91; *p* = 0.0002), and a subsequent Bonferroni test showed a significant increase in the ipsilateral side when compared to the naïve group (*p* = 0.0005) and the contralateral side (*p* = 0.0005). To finish, for Vimentin mRNA levels, one-way ANOVA showed a significant effect of lesion (F(2, 15) = 3.903; *p* = 0.0432) and Bonferroni post hoc test confirmed an increase in the ipsilateral side in comparison with the naïve animals (*p* = 0.0475).

To further deepen the study of the inflammatory profile in WT animals, four pro-inflammatory cytokines were evaluated (Figure 1g). Kruskal–Wallis showed a significant effect of lesion in Ccl5 mRNA levels (*p* = 0.039) and Mann–Whitney revealed an increase in the ipsilateral side in comparison with naïve (*p* = 0.030) and a trend when compared to the contralateral side (*p* = 0.052). Tnfa, Il6, and Ccl2 showed very similar expression changes. Kruskal–Wallis showed a significant effect of lesion on Tnfa (*p* = 0.001), Il6 (*p* = 0.004), and Ccl2 (*p* = 0.003). Mann–Whitney tests revealed an increase in Tnfa in the contralateral (*p* = 0.010) and ipsilateral (*p* = 0.004) in comparison with naïve mice and a difference between the two sides of the injured brain (*p* = 0.010). Similarly, Il6 presented a significant increase in the ipsilateral side when compared with the naïve (*p* = 0.009) and the contralateral side (*p* = 0.008) and Ccl2 showed a very high increase in the contralateral (*p* = 0.028) and the ipsilateral side in comparison with naïve (*p* = 0.006) and a difference between the two brain sides of the injured animals (*p* = 0.016). Taken together, these findings indicate that Nlrp3 and Aim2 inflammasomes are overexpressed in the ipsilateral side of the lesion, resulting in the induction of Il1b, elevated microglia, and astrocyte reactivity markers and high expression of pro-inflammatory cytokines in both sides of the lesion.

### 3.2. Il1β mRNA Expression and Protein Release Are Equal in WT and NLRP3 KO Mice

After analyzing the inflammasome status and inflammatory profile in WT mice, the next step was to study the IL1β pathway at mRNA and protein levels in WT and NLRP3 KO mice 24 h after the lesion (*n* = 6–11). The expression patterns of Il1b mRNA and protein were similar in WT and KO mice; they increased at 24 h in the ipsilateral side with no genotype-related differences and without the involvement of other inflammasomes. Three lesion conditions (naïve, contralateral, and ipsilateral) were analyzed in two genotypes (WT and NLRP3 KO). Since the data were not normally distributed, Kruskal–Wallis and Mann–Whitney U-tests were run for Il1b mRNA expression. Kruskal–Wallis showed an effect of lesion in WT animals (*p* = 0.004) and NLRP3 KO (*p* = 0.000), with a near-significant difference (*p* = 0.057) on the ipsilateral sides of WT and KO mice (Figure 2a, left). Subsequent Mann–Whitney U-tests revealed a significant increase in Il1b mRNA in the ipsilateral side of WT animals in comparison with the naïve (*p* = 0.005) and the contralateral side (*p* = 0.028). Within the NLRP3 KO group, Il1b mRNA levels were significantly increased in the ipsilateral side in comparison with the naïve (*p* = 0.000) and contralateral side (*p* = 0.000) of KO mice. Similarly, IL1β protein expression was analyzed for both genotypes and lesion conditions (*n* = 8–10) (Figure 2a, right). Kruskal–Wallis test showed a significant effect of lesion side in WT (*p* = 0.037) and NLRP3 KO (*p* = 0.006). Mann–Whitney test showed a significant increase in IL1β protein expression in the ipsilateral side of the WT compared to naïve animals (*p* = 0.003), and the same occurred within the KO group (*p* = 0.006).

Basal Nlrp3 mRNA levels were assessed together with Aim2 expression to examine whether there were compensatory mechanisms of other inflammasomes to induce IL1β signaling. Two-way ANOVA showed significant effects of genotype (F(1, 55) = 17.746; *p* = 0.000), lesion (F(2, 112) = 45.498; *p* = 0.000), and an interaction genotype*lesion (F(2, 112) = 16.104; *p* = 0.000). Data were non-homoscedastic, so a Games–Howell post hoc test was run, revealing a significant decrease in Nlrp3 mRNA levels in naïve KO mice in comparison with naïve WT (*p* = 0.005) and a significant increase in the ipsilateral side of KO animals when compared with the naïve group of this same genotype (*p* = 0.000) (Figure 2b, left). For Aim2 mRNA levels, two-way ANOVA showed a significant effect of lesion (F(2, 112) = 8.497; *p* = 0.001) that was confirmed by Bonferroni’s post hoc test as a significant increase in the ipsilateral side of WT animals (*p* = 0.037) (Figure 2b, right). These findings suggest an alternative pathway or mechanism to release IL1β, independent of NLRP3 or others inflammasome activation.

### 3.3. NLRP3 KO Mice Showed an Exaggerated Inflammatory Response after TBI

To analyze the response of NLRP3 KO mice to TBI, specific panels of microglia (Figure 3a) and astrocyte markers (Figure 3b) were run, together with pro-inflammatory cytokines (Figure 3c) (*n* = 6–11). Generally speaking, the astrocytic markers and the expression of pro-inflammatory cytokines on the ipsilateral side of KO animals showed an exaggerated response to TBI in comparison with WT mice (+symbol). Regarding microglia markers, two-way ANOVA showed a significant effect of lesion (F(2, 114) = 20.509; *p* = 0.000) in Cd11b mRNA levels. Subsequent Bonferroni’s post hoc test revealed an increase in the ipsilateral side of KO animals when compared with the contralateral (*p* = 0.001) and the naïve group (*p* = 0.000). The Clec7a microglial marker presented a similar pattern and Kruskal–Wallis test showed a significant effect of lesion in WT (*p* = 0.007) and KO (0.003) mice; Mann–Whitney U-test confirmed a significant increase in the ipsilateral side of WT animals (*p* = 0.002) and an almost significant difference in the contralateral side with the WT naïve group (*p* = 0.058). Within the KO group, there was a significant increase in the ipsilateral side when compared to the contralateral (*p* = 0.009) and the naïve group (*p* = 0.002). Non-parametric tests were run for the analyses of Tyrobp mRNA expression. Kruskal–Wallis test revealed a significant difference at basal levels (naïve) between WT and KO mice (*p* = 0.034); subsequent Mann–Whitney U-test showed a significant increase in the contralateral (*p* = 0.030) and the ipsilateral (*p* = 0.002) side of WT animals, whereas no differences were found in the NLRP3 KO mice. To complete this microglial panel, Trem2 expression was also analyzed, without finding significant effects of lesion or genotype.

One of the main goals of this work was to study the effect of microglia response on astrocyte function after a TBI. To do so, astrocytic markers were also run (Figure 3b). Gfap mRNA levels were analyzed by non-parametric test and Kruskal–Wallis showed a significant effect of lesion in WT (*p* = 0.013) and KO (*p* = 0.000), with a strong trend towards higher basal levels in KO than in WT (*p* = 0.059). Mann–Whitney test confirmed a significant increase in Gfap in the ipsilateral side of WT mice (*p* = 0.009) in comparison with the naïve group. Within the KO mice, there were significant differences between naïve and contralateral side (*p* = 0.000), naïve and ipsilateral (*p* = 0.000), and between contralateral and ipsilateral sides themselves (*p* = 0.001). The receptor Il1r1 showed a significant effect of lesion, only in the KO group (*p* = 0.000), which was confirmed by Mann–Whitney test, revealing a significant increase in Il1r1 in the ipsilateral side of KO mice when compared with the naïve (*p* = 0.002), the contralateral side (*p* = 0.000), and with the ipsilateral side of the WT genotype (*p* = 0.002), supporting the idea of an exaggerated response in the NLRP3 KO mice. Similar results were found for Steap4 mRNA levels, as Kruskal–Wallis revealed a significant effect of lesion exclusively in KO mice (*p* = 0.000). Mann–Whitney test found a significant increase in Steap4 mRNA levels in the ipsilateral side of WT mice when compared with their naïve group (0.034) and bigger effects within the KO strain, in which there were significant increases in the ipsilateral side in comparison with the naïve (*p* = 0.000), the contralateral side (*p* = 0.001), and with the ipsilateral side of WT animals (*p* = 0.001), together with a significant difference between the contralateral side and the naïve group (*p* = 0.000). 

To further analyze the possible interaction between these two types of glial cells, the mRNA levels of different cytokines were also included (Figure 3c). All the measured cytokines showed an exaggerated response in the ipsilateral side of the KO mice. Tnfa data were analyzed by Kruskal–Wallis non-parametric test and showed a significant effect of the lesion in both strains (WT *p* = 0.008 and KO *p* = 0.000) and a significant effect of genotype in the ipsilateral side (*p* = 0.002). A subsequent Mann–Whitney test confirmed an increase in the contralateral (*p* = 0.020) and ipsilateral sides (*p* = 0.005) of WT mice in comparison with the naïve group of the same genotype; similar increases were found in the contralateral (*p* = 0.003) and ipsilateral (*p* = 0.000) sides of KO animals when compared with the naïve KO group. In NLRP3 KO, there was also a significant increase in Tnfa mRNA levels in the ipsilateral side compared to the contralateral side (*p* = 0.000) and an exaggerated response of the ipsilateral side of KO animals compared to their WT counterparts (*p* = 0.002). Il6 mRNA levels showed a significant effect of the lesion in WT (*p* = 0.019) that was higher in KO animals (*p* = 0.000). Furthermore, a significant effect of genotype was found in the ipsilateral side (*p* = 0.002). Mann–Whitney test revealed a significant increase in Il6 mRNA expression in the ipsilateral side of WT mice in comparison with the naïve group (*p* = 0.018) and with the contralateral side (*p* = 0.018). Within the KO group, there was a significant increment in the ipsilateral side compared to the naïve (*p* = 0.000), contralateral (*p* = 0.000), and the ipsilateral side of the WT group (*p* = 0.002). Following this same trend, Ccl2 mRNA expression showed a strong effect of lesion in the WT group (*p* = 0.002) and the KO mice (*p* = 0.000), together with a significant effect of genotype in naïve (*p* = 0.013), contralateral (*p* = 0.019), and ipsilateral (*p* = 0.009), demonstrating that this cytokine was a key molecule in the response after TBI and that, similarly to Tnfa and Il6, presented an exaggerated response in the ipsilateral side of the KO animals in comparison with the WT groups. These results corroborate the pro-inflammatory profile found in WT animals 24 h after TBI and show that NLRP3 KO mice respond in an exaggerated manner to brain damage, as reflected by the elevation in the microglial and astrocytic markers and the pro-inflammatory cytokines. A brief color-code summary is presented in Figure 3d. 

### 3.4. Activation of NLRP3 Is Critical for Blood–Brain Barrier Integrity and Neurological Damage

The presence and the activation time of the NLRP3 inflammasome is important for maintaining the integrity of the blood–brain barrier (BBB) and the severity of the neurological damage that occurs after a brain trauma. NLRP3 KO animals pre-treated with the selective inhibitor MCC950 showed an impaired BBB integrity and more neurological damage; these results were improved when MCC950 was given one hour after trauma. BBB rupture was assessed by the amount of the Evans Blue tracer that was extravasated (*n* = 8–13). The genetic approach to compare WT and NLRP3 KO BBB integrity (Figure 4a,c) showed a significant increase in BBB rupture in KO animals compared to WT (unpaired *t*-test, *t* = 2.104; df = 21; *p* = 0.0476). A pharmacological approach using a selective inhibitor of NLRP3 inflammasome (MCC950, 3 mg/kg) was also included (Figure 4b,c). One-way ANOVA showed a significant effect of treatment (pre, vh, post) (F(2, 29) = 8.341; *p* = 0.0014). A subsequent Bonferroni test confirmed a significant increase in the BBB rupture in animals pre-treated with MCC950 compared to vh animals (*p* = 0.0423), a trend to be decreased by the post-treatment (*p* = 0.082) and a significant difference between pre- and post-treated animals (*p* = 0.0012). These findings were supported by behavioral tests. The neurological score (NSS) test was performed at 1 h and 24 h after TBI to find differences between the strains (Figure 4d). Kruskal–Wallis showed a significant effect of time (1 h, 24 h) in WT animals (*p* = 0.034), whereas KO mice did not improve their neurological score at 24 h. Similar results were obtained when using the pharmacological approach (vh, pre, post), inhibiting the NLRP3 inflammasome 30 min prior to the TBI (pre) or 1 h after injury (post) (Figure 4e). Kruskal–Wallis revealed a significant effect of time (1 h, 24 h) in vehicle-treated mice (*p* = 0.034) and a bigger effect in the post-treated group (*p* = 0.001). A subsequent Mann–Whitney test showed a significant decrease in the neurological damage of post-treated animals compared to vehicle-treated (*p* = 0.038) and pre-treated mice (*p* = 0.006). Taken together, these findings highlight the importance of NLRP3 in maintaining BBB integrity and, thus, affecting neurological damage and the dual time-dependent role of this inflammasome to induce brain repair. In addition, the treatment with MCC950 1 h after TBI reduced the BBB rupture and the neurological damage, suggesting that this selective NLRP3 inhibitor could be a potential treatment for TBI patients if administered within the proper temporal window. 

## 4. Discussion

Increasing evidence suggests that inflammasomes play an important role in the regulation of post-TBI neuroinflammatory responses. The present study reports the importance of NLRP3, the canonical sensor of sterile injury, in the microglia and astrocyte responses during the acute phase of brain trauma. Consistent with the bibliography [9,13,19,20,21], we showed that traumatic brain injury induces an inflammatory response in WT animals, increasing the mRNA levels of glial pro-inflammatory markers and cytokines. In addition, NLRP3 KO animals have been shown to have an exaggerated post-TBI inflammatory response, with increased astrocyte reactivity markers and astrocyte-related cytokines, probably due to a dysfunctional signaling interaction between microglia and astrocytes. This exaggerated inflammatory response is detrimental because the absence of NLRP3 in knockout mice and the selective inhibition with MCC950 before brain trauma resulted in increased BBB permeability, leading to total BBB loss and an exacerbation of the neurological damage. However, animals treated with MCC950 1 h after brain trauma showed a better BBB integrity and, thus, a better neurological outcome. These data indicate that NLRP3 plays a dual time-dependent role in the acute phase of TBI, supporting the idea of a critical role for NLRP3 in the inflammatory response of microglia and astrocytes and corroborating the impact of its activation on BBB integrity and the severity of the trauma.

As mentioned in the Introduction, neuroinflammation plays a central role in the secondary phase of TBI, causing cellular degeneration and changes in neural/synaptic transmission and plasticity [22]. However, different inflammasomes are activated during the acute phase of TBI. The inflammasomes share a similar structure and are typically composed of a cytosolic pattern-recognition receptor, an adaptor protein, and an effector component (caspase-1 and -11). Within the known inflammasomes, NLRP3 has been reported to be associated with neuroinflammation in the pathological onset of neurodegenerative diseases and secondary inflammation following TBI [9,13,20,23]. Various authors have shown that at 24 h after trauma, there is a peak in the expression of different inflammasomes, NLRP1, AIM2, NLRP3, and some of their components [11,23,24]. We showed an increased expression of Aim2 and Nlrp3 mRNA levels with increased Il1b (mRNA and protein) in WT animals. However, no increase in Nlrp1, Nlrc4, and other inflammasome components, such as ASC, Casp1, or Il18, was observed, whereas other studies found significant changes in these transcripts [9,11]. A possible explanation could be the specific animal model that was used to perform the brain trauma. In this study, we used a severe non-repetitive TBI (closed-head injury) murine model, whereas the above-mentioned works refer to an open-head model, performed with rats or a controlled cortical impact in mice. Because TBI is a complex pathophysiological event, there are different types of TBI models that allow us to study different features of brain injury. All the current animal models have their strengths and weaknesses, and some are closer to the conditions that are observed in humans. Our weight-drop model duplicates, with high fidelity, the concussion, contusion, axonal injury, and hemorrhage observed in patients and also has the advantage of assessing the neurological damage as soon as 1 h [25]. The genetical approach, comparing WT and NLRP3 KO mice, showed a similar increase in Il1b expression (mRNA and protein) in both genotypes, without any change in the expression of other inflammasomes (Aim2, Nlrc4). These findings are consistent with various studies showing that IL1β signaling is important for neuroinflammation after a TBI but not necessarily associated with any inflammasome activation (NLRP3 or other) [23] or the canonical caspase 1 and 11 cleavage, suggesting an alternative pathway for IL-1β maturation, such as caspase-8, neutrophil proteases [26], elastase, chymase, or proteinase 3 [27]. It is known that caspase-8 processes IL1β using the same site targeted by caspase-1 [28,29]. Experiments with dendritic cells exposed to fungi showed the formation of a non-canonical caspase-8 inflammasome that matures IL1β through dectin-1 signaling [30] and in vivo TBI models demonstrated that the genetic deletion of caspase-8 improved the functional and histopathological outcomes after TBI [31]. These studies suggest that, in the absence or inhibition of the canonical NLRP3 pathway, the apoptotic pathway takes over and caspase-8, as the main initiator of it, leads the response after brain damage. In fact, a temporal profile of caspase-8 expression in a controlled cortical impact in rats has been described, showing increased mRNA levels from 1 h to 72 h after injury [32]. More recent studies in TBI patients showed that high serum caspase-8 concentrations are associated with mortality [33]. However, more experiments are needed to confirm that these same signaling pathways are activated in a murine model of close-head injury as ours (see also [34] for review).

Microglia are one of the most important cell types that contribute to the inflammatory response after brain trauma. Depending on their activation status and phenotype, they have different functions under normal health conditions [35]. Homeostatic microglia have been associated with immunoprotection, synaptic modeling, and neuronal health support, and the reactive dysfunctional microglia present impaired phagocytic capacity, loss of homeostatic function, and a pro-inflammatory profile. Hickman et al. identified 100 transcripts highly enriched in microglia by using RNA sequencing and coined the term “sensome” to describe the molecular signature of homeostatic microglia [36] and, more recently, Keren-Shaul et al., using single-cell RNA sequencing, defined a specific microglial phenotype that the authors termed “disease-associated microglia” and proposed that this state is acquired through a two-step mechanism of activation of homeostatic microglia [37]. An initial activation of microglia leads to an intermediate state (stage 1) in a Trem2-independent mechanism, which involves the upregulation of genes, such as B2m, Apoe, and Tyrobp, and this state can be further activated by a second signal that is Trem2-dependent (stage 2) and involves the upregulation of phagocytic genes, such as Clec7a, Cd11c, or Trem2 [34]. Our results show that Trem2 mRNA levels did not change in any group at 24 h after trauma, suggesting that microglia could be at this first Trem2-independent activation stage. However, Tyrobp was significantly upregulated in naïve NLRP3 KO mice. To our knowledge, here, we demonstrate, for the first time, that NLRP3 KO animals present alterations in Tyrobp mRNA at basal conditions. Tyrobp (tyrosine kinase binding protein, also known as DAP12, for DNAX activating protein-12) has been described as one of the most upregulated genes involved in the Trem2-independent transition from homeostatic microglia to the activated state [38,39]. The fact that Tyrobp is elevated in NLRP3 KO mice suggests the idea that microglial “sensome” is altered, is more responsive to damage or inflammatory insults, and could present some characteristics of “primed” microglia. Consistent with this reasoning, WT animals showed increased Tnfa mRNA expression (microglial response) and NLRP3 KO animals presented an enhanced and exaggerated response to the same lesion. The factors and sequential order that facilitate the transition from homeostatic microglia to the activated state are currently unknown. Therefore, we hypothesize that the NLRP3 inflammasome may be one of the mediators of this transition and the absence of it might alter the microglial response and the subsequent sequence of events. 

Interactions between different sets of glial cells in CNS neuroinflammation are highly complex and dynamic and their communication is crucial to produce an adequate response to different insults. As previously mentioned, microglia are the first cell type to sense changes in the brain and to respond accordingly. The type of response that they start and the way they communicate with the surrounding cells, particularly astrocytes, determine the fate and the outcome of the damage [40,41,42]. Astrocytes respond to a variety of CNS injuries and diseases [43,44] and are important for their role in synaptic transmission, plasticity, and neurovascular functions [45,46,47,48,49,50,51]. Recently, we showed that reducing the astrocytic TLR4 receptor activation and the pro-inflammatory microglial phenotype within the first few hours after TBI can promote synaptic plasticity and improve cognitive function through astrocytic modulation [52]. Activation of TLR4 is the first step in the two signals required for the NLRP3 inflammasome activation [53,54]. Here, we show that primed microglia of NLRP3 KO animals resulted in an exaggerated response of astrocytes to TBI compared to WT animals, showing an enhanced response of the Il1r1 and Steap4 genes (astrocytic markers) that lead to an overexpression of cytokines (Il6 and Ccl2, together with Tnfa). Evidence on physical contacts between these cells is not yet available, but it is known that there is an exchange of trophic factors, ATP, and amino acids through P2Y receptors [55], aquaporin-4 [56], or extracellular vesicles [57], and here, we hypothesize that the NLRP3 signaling pathway may also be involved in astrocyte reactivity, depending on an initial functional or dysfunctional microglia response in health and disease conditions.

However, in contrast to the finding of primed microglia in naïve KO animals, no primed astrocyte characteristics were found since there was not a basal increase in the expression of reactive astrocyte markers. Therefore, it seems that there is a dysregulation in the microglia homeostatic state, which also appears to affect the function of astrocytes. In addition, this dysfunction of cell–cell interaction also results in immediate BBB breakdown, probably due to the release of astrocyte proteins and pro-inflammatory cytokines (Ccl2, Il6) known to have detrimental effects on the BBB [58,59,60,61]. NLRP3 seems critical for proper inflammatory response and control of the BBB permeability, as NLRP3 KO animals and WT animals treated with the selective NLRP3 inhibitor MCC950 (3 mg/kg; i.p.) prior to TBI showed increased BBB rupture compared to WT TBI animals. However, once NLRP3 is activated, this inflammatory response should be selectively controlled to observe a protective effect in BBB integrity. For this reason, MCC950 at the same dose (3 mg/kg; i.p.) exerted a protective effect when administered 1 h after TBI, resulting in better BBB integrity and less neurological damage. These findings are consistent with other studies using selective NLRP3 inhibitors [12,13,20] and with our previous results in a temporal brain ischemia murine model [14]. These data confirm that the activation of NLRP3 is crucial for a proper microglia response and the subsequent astrocyte reaction to produce a good damage recovery after TBI. However, more experiments are needed to study the time course, the molecular mechanisms underpinning this crosstalk, and to determine whether there is direct microglia–astrocyte communication that is strictly initiated by the inflammasome. 

In summary, this study shows, for the first time, that NLRP3 KO mice have some primed microglial characteristics, in which elevated Tyrobp mRNA levels may be important, leading to exaggerated astrocyte response and cytokine expression after TBI. The fact that the genetic ablation of NLRP3 and its pharmacological inhibition impairs BBB integrity and neurological damage, together with a better recovery when the inhibitor was administered after the lesion, emphasizes the time-dependent and fine-tuning regulation of NLRP3 to induce a proper inflammatory response and damage recovery. These results also open the possibility of using MCC950 as a promising approach for TBI patients when the therapeutic window is optimal.

## 5. Conclusions

The NLRP3 inflammasome is key for a proper microglia early response and may affect the function of astrocytes to induce a functional and beneficial inflammatory response after TBI.

## Figures and Tables

**Figure 1 pharmaceutics-14-01550-f001:**
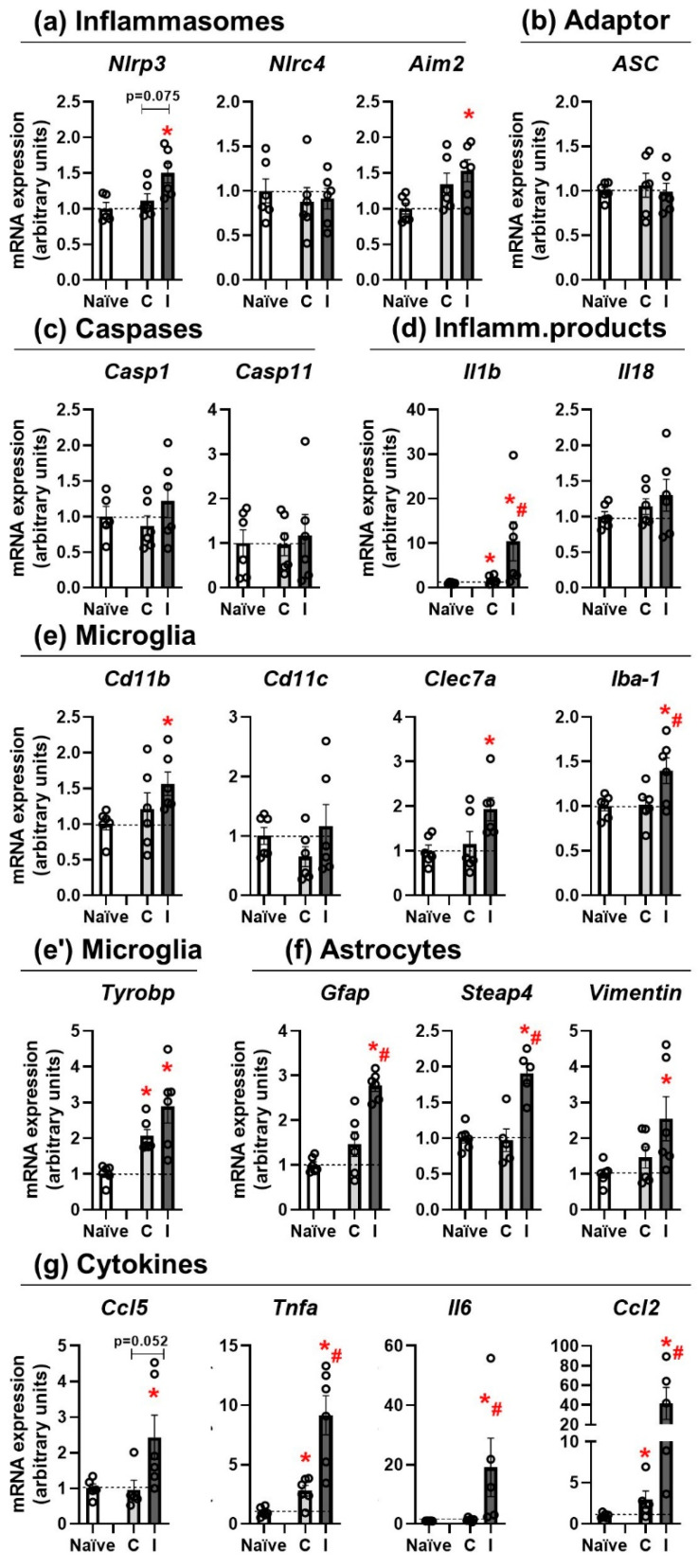
Inflammatory profile in WT 24 h after TBI. (**a**) Effects of lesion in inflammasome transcript expression. (**b**) Changes in the adaptor protein ASC. (**c**) Caspase 1 and 11 mRNA levels 24 after TBI. (**d**) Inflammasome product changes after lesion. (**e/e′**) Microglia activation status after injury. (**f**) Activation markers of astrocytes 24 h after TBI. (**g**) Pro-inflammatory cytokine mRNA expression in naïve and injured animals. Mean ± SEM (*n* = 6). Kruskal–Wallis or one-way ANOVA followed by Mann–Whitney or Bonferroni’s tests *p* ≤ 0.05. (* vs. Naïve; # vs. Contra). (C: contralateral; I: ipsilateral).

**Figure 2 pharmaceutics-14-01550-f002:**
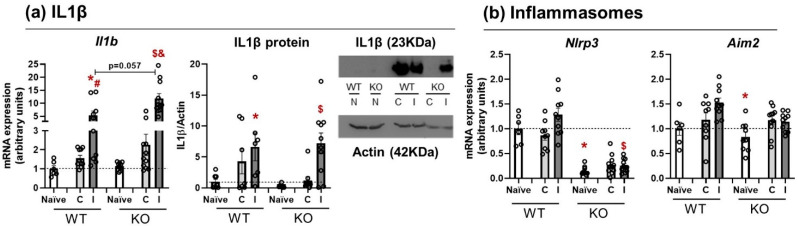
IL1β signaling in WT and NLRP3 KO mice 24 h after the lesion. (**a**) Changes in Il1β mRNA (**left**) and protein expression (**center**) in WT and KO mice. Naïve, contralateral, and ipsilateral sides were compared in both strains. Representative Western blot image of IL1β protein and actin is shown (**right**). (**b**) Changes in Nlrp3 and Aim2 mRNA levels in WT and KO. Naïve, contralateral, and ipsilateral sides were compared in both strains. Mean ± SEM (*n* = 6–11). Kruskal–Wallis or two-way ANOVA followed by Mann–Whitney or Bonferroni’s tests *p* ≤ 0.05. (* vs. WT Naïve; # vs. WT Contra; $ vs. KO Naïve; & vs. WT Ipsi (WT: wild type; KO: NLRP3 knockout; C: contralateral; I: ipsilateral).

**Figure 3 pharmaceutics-14-01550-f003:**
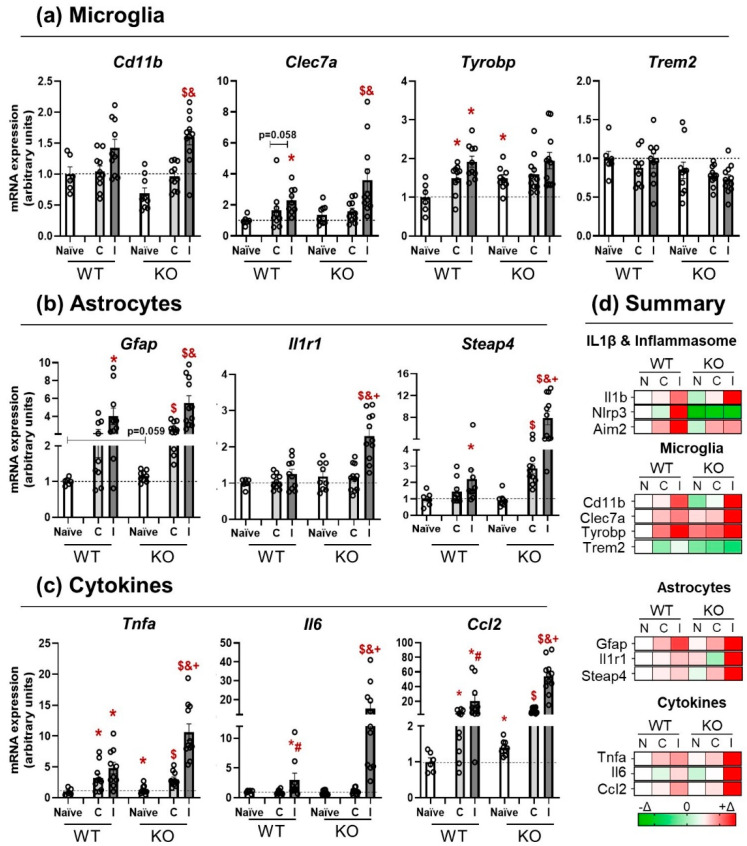
Glial response and cytokine expression in WT and KO mice 24 h after trauma. (**a**) Microglia transcripts panel, showing the changes in the mRNA levels of Cd11b, Clec7a, Tyrobp, and Trem2. Naïve, contralateral, and ipsilateral sides were compared in both strains. (**b**) Response of the astrocytic markers Gfap, Il1r1, and Steap4 in WT and KO. Naïve, contralateral, and ipsilateral sides were compared in both strains. (**c**) mRNA expression of the pro-inflammatory cytokines Tnfa, Il6, and Ccl2 in WT and KO animals. Naïve, contralateral, and ipsilateral sides were compared in both strains. (**d**) Graphical summary of the changes observed at mRNA level for all the studied transcripts. Green colors represent a decrease, white means no change, and red palette corresponds to increases in the mRNA expression. Mean ± SEM (*n* = 6–11). Kruskal–Wallis followed by Mann–Whitney tests *p* ≤ 0.05. (* vs. WT Naïve; # vs. WT Contra; $ vs. KO Naïve; &: KO Ipsi vs. KO contra; + KO Ipsi vs. WT Ipsi (WT: wild type; KO: NLRP3 knockout; C: contralateral; I: ipsilateral).

**Figure 4 pharmaceutics-14-01550-f004:**
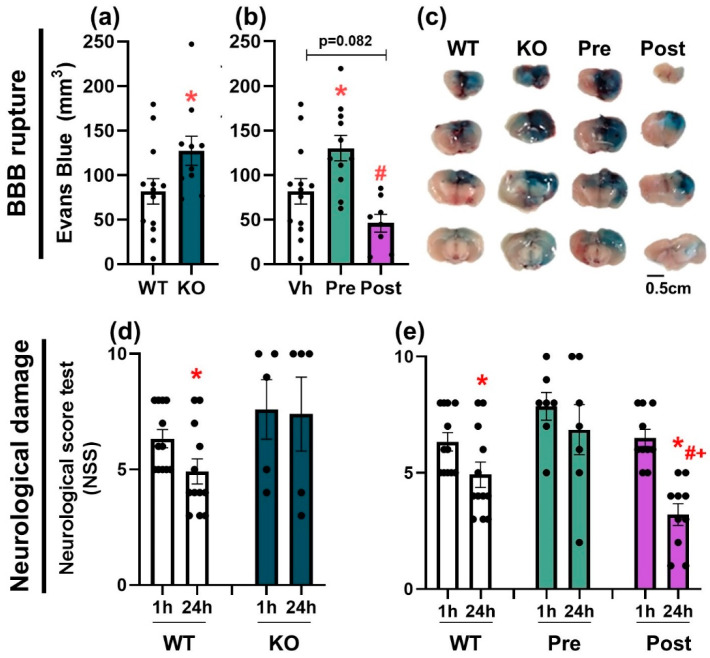
Genetic and pharmacological approach for NLRP3 inactivation. (**a**) BBB rupture in WT and KO mice 24 h after trauma. Mean ± SEM (*n* = 10–13). (**b**) BBB rupture in vh-, preMCC950-, and post-MCC950-treated mice. Mean ± SEM (*n* = 8–13). Kruskal–Wallis or one-way ANOVA followed by Mann–Whitney or Bonferroni’s tests *p* ≤ 0.05. (* vs. WT (Vh); # vs. Pre). (**c**) Representative pictures of Evans Blue tracer extravasation in WT (vh-treated), KO, Pre-, and Post-MCC950 animals. (**d**) NSS test in WT and KO mice at 1 and 24 h after trauma. Mean ± SEM (*n* = 5–12). (**e**) NSS test in vh-, pre-MCC950-, and post-MCC950-treated mice at 1 and 24 h after injury. Mean ± SEM (*n* = 7–12). Kruskal–Wallis followed by Mann–Whitney test *p* ≤ 0.05. (* vs. 1 h within the same treatment; # vs. Vh-24 h; + vs. Pre-24 h).

**Table 1 pharmaceutics-14-01550-t001:** Neurological Severity Score test.

Task	Description
Exit circle	Exit a circle of 30 cm od diameter in 2 min
Seeking behavior	Interest on the environment and movement
Motor skills	Straight walk Monoparesis or hemiparesis
Startle reflex	Response to a loud hand clap
Beam balancing	Ability to balance on a surface of 10 × 10 mm
Beam walk	Cross a 30 cm-long beam of 3 cm-width Cross a 30 cm-long beam of 2 cm-width Cross a 30 cm-long beam of 1 cm-width
Round stick balancing	Ability to balance on a 5 mm round stick

Quotation for each test and assessed task: success in each task is considered 0 points; failure is considered 1 point.

**Table 2 pharmaceutics-14-01550-t002:** Primer sequences.

Gen	Primer Sequences
18S	Fw: 5′-CGCCGCTAGAGGTGAAATTCT-3′ Rv: 5′-CATTCTTGGCAAATGTCTTTCG-3′
Aim2	Fw: 5′-AAGAGAGCCAGGGAAACTCC-3′ Rv: 5′-CACCTCCATTGTCCCTGTTT-3′
ASC	Fw: 5′-GAAGCTGCTGACAGTGCAAC-3′ Rv: 5′-GAAGAGTCTGGAGCTGTGGC-3′
Casp1	Fw: 5′-AGATGGCACATTTCCAGGAC-3′ Rv: 5′-GATCCTCCAGCAGCAACTTC-3′
Casp11	Fw: 5′-ACGATGTGGTGGTGAAAGAGGAGC-3′ Rv: 5′-TGTCTCGGTAGGACAAGTGATGTGG-3′
Ccl2	Fw: 5′-ACAAGAGGATCACCAGCAGC-3′ Rv: 5′-GGACCCATTCCTTCTTGGGG-3′
Ccl5	Fw: 5′-GCAGTCGTGTTTGTCACTCGAA-3′ Rv: 5′-GATGTATTCTTGAACCCACTTCTTCTC-3′
Cd11b	Fw: 5′-CCTTGTTCTCTTTGATGCAG-3′ Rv: 5′-GTGATGACAACTAGGATCTT-3′
Cd11c	Fw: 5′-CTGGATAGCCTTTCTTCTGCTG-3′ Rv: 5′-GCACACTGTGTCCGAACTC-3′
Clec7a	Fw: 5′-CCCAACTCGTTTCAAGTCAG-3′ Rv: 5′-AGACCTCTGATCCATGAATCC-3′
Gfap	Fw: 5′-CTCCAACCTCCAGATCCGAG-3′ Rv: 5′-TCCACAGTCTTTACCAGATGT-3′
Iba1	Fw: 5′-CCGAGGAGACGTTCAGCTAC-3′ Rv: 5′-GACATCCACCTCCAATCAGG-3′
Il18	Fw: 5′-GACTCTTGCGTCAACTTCAAGG-3′ Rv: 5′-CAGGCTGTCTTTTGTCAACGA-3′
Il1b	Fw: 5′-AACCTGCTGGTGTGTGACGTTC-3′ Rv: 5′-CAGCACGAGGCTTTTTTGTTGT-3′
Il1r1	Fw: 5′-GCAATATCCGGTCACACGAGTA-3′ Rv: 5′-ATCATTGATCCTGGGTCAGCTT-3′ Probe: 5′-TCCTGAGCCCTCGGAATGAGACGATC-3′
Il6	Fw: 5′-TTCTCTGGGAAATCGTGGAAA-3′ Rv: 5′-CTGCAAGTGCATCATCGTTGT-3′
Nlrc4	Fw: 5′-TGGTGACAATAGGGCTCCTC-3′ Rv: 5′-CTGTTCCCTTTGCTCACCTC-3′
Nlrp3	Fw: 5′-GCCCAAGGAGGAAGAAGAAG-3′ Rv: 5′-TCCGGTTGGTGCTTAGACTT-3′
Steap4	Fw: 5′-TGCAAGCCGGCAGGTGTTTGT-3′ Rv: 5′-TCCAGTGGGGTGAGCCCAAGA-3′
Tnfa	Fw: 5′-GCCTCTTCTCATTCCTGCTTG-3′ Rv: 5′-CTGATGAGAGGGAGGCCATT-3′
Tyrobp	Fw: 5′-CGTACAGGCCCAGAGTGAC-3′ Rv: 5′-CACCAAGTCACCCAGAACAA-3′
Vimentin	Fw: 5′-GCTGCAGGCCCAGATTCA-3′ Rv: 5′-TTCATACTGCTGGCGCACAT-3′

Where no probe sequence is shown, SYBR green dye was used instead. Fw: Forward; Rv: reverse.

## Data Availability

Available on demand.

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
