# Peer review of "Activation of NLRP3 Is Required for a Functional and Beneficial Microglia Response after Brain Trauma"

_pharmaceutics, 2022, doi:10.3390/pharmaceutics14081550_

Round 1

Reviewer 1 Report

This manuscript entitled “NLRP3 activation is required for proper microglia-astrocyte 2 communication after brain trauma.” investigated how NLRP3 inflammasome activation alters blood brain barrier (BBB) permeability following brain trauma.  The authors used both a genetic and pharmacological inhibition of NLRP3 to investigate how the NLRP3 inflammasome alters TBI injury. Overall the authors data suggest that specific inhibition of NLRP3 following TBI may be beneficial to limit BBB permeability, whereas inhibition of NLRP3 prior to TBI may increase BBB permeability, implicating NLRP3 activation as a potential of microglia and astrocyte regulator.

Below are a few concerns for the authors to improve their manuscript.

Concerns

·         Please confirm all abbreviations used throughout the manuscript (including figures). At times the authors use the alpha symbol and at other times use the letter “a”. Please use the proper format throughout the manuscript.   

·         Was the investigator rating the mice using the NSS test blinded from the different groups?

Reviewer 2 Report

The paper by Belen Lopez-Rodriguez et al., provides interesting data that may explain why several other reports in the field show NLRP3 KO animals do not show protection in a variety of CNS injuries. Their main finding is that the loss or blockade of NLRP3 in the acute phase (under 1h) is detrimental to recovery after TBI, but blockade after 1h is beneficial, suggesting extraordinary temporal effects of the inflammatory response.   

The experiments appear to have been done rigorously and is therefore an important addition to the field.

There are, however, several instances of over interpretation, which is unnecessary.

This includes the idea that the data provide evidence for microglia-astrocyte communication. This should be removed from the title and, at most, speculated at in the discussion.  

To suggest this, it would require that the authors specifically manipulate one of the two cells and measure the response in the other. They suggest that as ‘microglia are one of the most important cell types that contribute to the inflammatory response after brain trauma’, NLRP3 KO preferentially affects microglia, then astrocytes. They also claim that increased expression of a single gene, Tyrobp, in KO vs WT naïve mice is sign of ‘primed’ microglia. To claim this, the authors either need to show much more evidence (other microglia specific gene changes or morphology for example), or tone down their conclusions.

In addition, the authors say they previously knocked out TLR4 from astrocytes, therefore reducing the signal for NLRP3 in astrocytes. Indeed, NLRP3 is not restricted to microglia, so their global KO or administration of MCC950 should not be assumed to affect microglia preferentially, with subsequent effects on astrocytes.

Secondly, slight increase of one gene, Tyrobp, does not really suggest these cells are similar to DAMs, especially as none of the other signature genes change. Personally, am not in favour of labels such as DAMs. The danger of these labels/signatures (described by scRNAseq) is that for those not doing scRNAseq begin to pick one gene (if it changes) and ascribe an entire phenotype or even function to the cell. As is the case here.

After this, I think the study throws up some very intriguing questions. As is shown by their IL-B protein expression (Fig 2a), the NLRP3 effect on BBB breakdown and neurological function is independent of IL-1B protein production. This is quite significant. Can the authors speculate as to what is happening of it’s not through IL-1B?

Do the authors have IL-B protein data (or any gene expression data) for the MCC950 treatment? This is not essential to publication, but would be a very nice addition.
